# Relating the topology of Dirac Hamiltonians to quantum geometry: When the quantum metric dictates Chern numbers and winding numbers

**Bruno Mera**[1,2,3*], **Anwei Zhang**[4,5] **and Nathan Goldman**[6†]

**1** Instituto de Telecomunicações, 1049-001 Lisboa, Portugal
**2** Departmento de Física, Instituto Superior Técnico, Universidade de Lisboa,
Av. Rovisco Pais, 1049-001 Lisboa, Portugal
**3** Departmento de Matemática, Instituto Superior Técnico, Universidade de Lisboa,
Av. Rovisco Pais, 1049-001 Lisboa, Portugal
**4** Department of Physics, The Chinese University of Hong Kong, Shattin,
New Territories, Hong Kong, China
**5** Department of Physics, Ajou University, Suwon 16499, Korea
**6** Center for Nonlinear Phenomena and Complex Systems, Université Libre de Bruxelles,
CP 231, Campus Plaine, B-1050 Brussels, Belgium

⋆ bruno.mera@tecnico.ulisboa.pt, † ngoldman@ulb.ac.be

## Abstract

Quantum geometry has emerged as a central and ubiquitous concept in quantum sciences, with direct consequences on quantum metrology and many-body quantum physics. In this context, two fundamental geometric quantities are known to play complementary roles: the Fubini-Study metric, which introduces a notion of distance between quantum states defined over a parameter space, and the Berry curvature associated with Berry-phase effects and topological band structures. In fact, recent studies have revealed direct relations between these two important quantities, suggesting that topological properties can, in special cases, be deduced from the quantum metric. In this work, we establish general and exact relations between the quantum metric and the topological invariants of generic Dirac Hamiltonians. In particular, we demonstrate that topological indices (Chern numbers or winding numbers) are bounded by the quantum volume determined by the quantum metric. Our theoretical framework, which builds on the Clifford algebra of Dirac matrices, is applicable to topological insulators and semimetals of arbitrary spatial dimensions, with or without chiral symmetry. This work clarifies the role of the Fubini-Study metric in topological states of matter, suggesting unexplored topological responses and metrological applications in a broad class of quantum-engineered systems.

## 1 Introduction

Recent advances have revealed the central role played by the Fubini-Study metric [1] in various fields of quantum sciences [2], with a direct impact on quantum technologies [3, 4] and many-body quantum physics [2, 5]. In condensed matter, the quantum metric generally defines a notion of distance over momentum space, and it was shown to provide essential geometric contributions to various phenomena, including exotic superconductivity [6–8] and superfluidity [9], orbital magnetism [10, 11], the stability of fractional quantum Hall states [12–17], semiclassical wavepacket dynamics [18, 19], topological phase transitions [20], and light-matter coupling in flat-band systems [21]. Besides, the quantum metric plays a central role in the construction of maximally-localized Wannier functions in crystals [22, 23], and it provides practical signatures for exotic momentum-space monopoles [24, 25] and entanglement in topological superconductors [26, 27].

    Motivated by these developments, the quantum metric was experimentally measured in several quantum-engineered systems, including cold atoms in optical lattices [28], NV centers in diamond [29–31], exciton polaritons [32] and superconducting qubits [33, 34]. The generalization of the quantum metric to mixed states (also known as the Bures metric) was also recently estimated through randomized measurements [35].

    In this quantum-geometry context, surprising connections have been made between the quantum metric and the Berry curvature of Bloch states. The latter captures Berry-phase effects in Bloch bands [36] and constitutes the central ingredient for the construction of topological invariants, such as the Chern number [37, 38]. For instance, in superconductors, the integral of the quantum metric over momentum space captures the superfluid weight in flat bands, which was found to be bounded from below by the Chern number [6]. In Weyl-type systems,

relations between the determinant of the quantum metric and the Berry curvature were shown to facilitate the observation of exotic topological defects based on quantum-metric measurements [24,25,31,34]. In systems of chiral multifold fermions, the trace of the quantum metric was shown to be quantized [39,40], and related to the Chern number through an intriguing sum rule involving the states' angular momentum. In the context of Chern insulators, relations between the Berry curvature, the Chern number, the quantum metric and the quantum volume were recently established and understood based on the Kähler structure of the quantum-states space [41,42]. These relations are known to play an important role in the formation and stabilization of fractional Chern insulators [12–17,43].

**Scope, main results and outline**

In this article, we establish general and exact relations between the quantum metric of generic Dirac Hamiltonians and the topological invariants of the corresponding Bloch bands, in arbitrary spatial dimensions $d$. This general framework, which builds on the Clifford algebra of Dirac matrices, expresses the topological indices of Chern insulators, but also chiral insulators and topological semimetals, in terms of the determinant of the quantum metric. These results highlight the central role played by the Fubini-Study metric in topological states of matter, but it also suggests unexplored topological responses in quantum-engineered settings. Indeed, synthetic lattice systems are currently developed in a broad range of experimental settings, including ultracold gases, photonics devices and electric circuits, in view of realizing ideal Dirac toy models of topological matter; emblematic examples of synthetic Dirac systems include the one-dimensional Su-Schrieffer-Heeger model for cold atoms using optical superlattices [44], the two-dimensional Haldane model in circularly-shaken honeycomb lattices [28,45,46], the ideal three-dimensional Weyl model realized in spin-orbit coupled gases [47], and the electric circuit realization of a four-dimensional topological Dirac model [48]. Interestingly, these synthetic lattice systems allow for momentum-resolved measurements of geometric properties, including the Berry curvature and the quantum metric [28,32,49,50].

Specifically, we first obtain general relations between the determinant of the quantum metric and two types of topological invariants: the $n$-th Chern character for systems without chiral symmetry in $2n$ spatial dimensions, and the winding-number class for systems with chiral symmetry in $2n-1$ spatial dimensions. These relations [Eqs. (15) and (34)], which are the central results of this work, stem from a special mapping between the system's Brillouin zone and a sphere, as we illustrate in Fig. 1. Upon integration over the Brillouin zone, these relations provide useful inequalities between topological indices of Bloch bands and the *quantum volume*: For gapped systems without chiral symmetry in $d = 2n$ dimensions, this inequality involves the $n$-th Chern number and reads

$$|\mathrm{Ch}_n| \leq \frac{(2n)!}{2^{n(n-1)+1}n!\pi^n}\mathrm{vol}_g(\mathbb{T}^{2n}), \tag{1}$$

where the quantum volume $\mathrm{vol}_g(\mathbb{T}^{2n})$ is defined as the integral of the quantum metric's determinant over the Brillouin zone, $\mathrm{vol}_g(\mathbb{T}^{2n}) = \int_{\mathbb{T}^{2n}}\sqrt{\det(g)}\,d^{2n}k$. A similar inequality is obtained for the winding number of $(2n-1)$-dimensional chiral insulators,

$$|\nu| \leq \frac{(n-1)!}{2^{\frac{1}{2}(n-1)(2n-5)}\pi^n}\mathrm{vol}_g(\mathbb{T}^{2n-1}). \tag{2}$$

The relations presented in Eqs. (1) and (2) show that the volume of the Brillouin zone, as measured by the quantum metric, provides an upper bound to the topological invariants of generic Dirac Hamiltonians, in all dimensions.



**(a) No chiral symmetry**

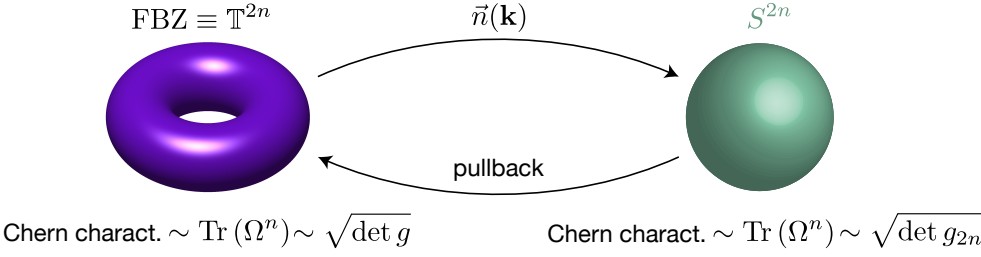

$\text{FBZ} \equiv \mathbb{T}^{2n}$ $\qquad \vec{n}(\mathbf{k}) \qquad$ $S^{2n}$

pullback

Chern charact. $\sim \text{Tr}\,(\Omega^n) \sim \sqrt{\det g}$ $\qquad$ Chern charact. $\sim \text{Tr}\,(\Omega^n) \sim \sqrt{\det g_{2n}}$

**(b) Chiral symmetry**

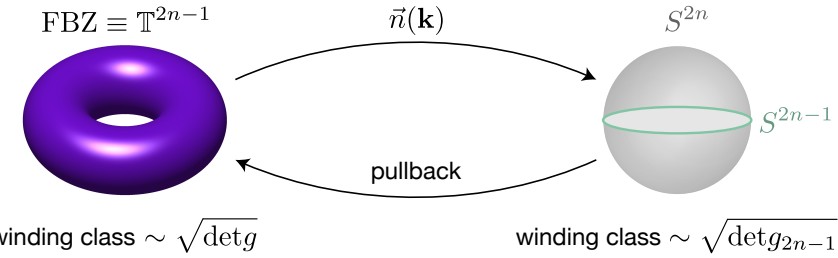

$\text{FBZ} \equiv \mathbb{T}^{2n-1}$ $\qquad \vec{n}(\mathbf{k}) \qquad$ $S^{2n}$

$S^{2n-1}$

pullback

winding class $\sim \sqrt{\det g}$ $\qquad$ winding class $\sim \sqrt{\det g_{2n-1}}$

Figure 1: Schematics of the main results: (a) In the case of gapped systems without chiral symmetry, in $d = 2n$ dimensions, the first Brillouin zone (FBZ) is mapped to the $2n$-sphere by $\vec{n}(\mathbf{k})$; see Eq. (5). The topological properties of Bloch states are then deduced from the topology of quantum states over the $2n$-sphere, which involves fundamental relations with the determinant of the metric [Eqs. (9) and (15)]. (b) A similar picture holds for chiral systems, where the map $\vec{n}(\mathbf{k})$ is now restricted to the equator $S^{2n-1}$ of $S^{2n}$, whose topology is captured by a winding-number class [Eqs. (13) and (34)]. The Dirac Hamiltonian involves $D = 2n + 1$ Dirac matrices in the non-chiral case (a) and $D = 2n$ Dirac matrices in the chiral case (b); see Eq. (3) and text below.

We describe below how these inequalities can be saturated, and we illustrate how they can be used in practice so as to deduce the topological nature of exotic quantum matter from quantum-metric measurements.

This article is organized as follows: Section 2 introduces the general framework of Dirac Hamiltonians, setting the focus on their geometric and topological properties using the language of differential forms. More explicit relations between the quantum metric, the components of the Berry curvature and the Chern numbers are then specified in Section 3 for the case of topological insulators without chiral symmetry. Chiral insulators, which are characterized by winding numbers, are then treated in Section 4. Topological semimetals are eventually discussed in Section 5. Each Section illustrates our general formula based on concrete and emblematic models of topological matter. We finally illustrate the implications of our metric-curvature relations for quantum metrology, where the quantum metric is known to quantify the metrological potential of quantum states via the Cramér-Rao bound [51, 52]. Our concluding remarks are presented in Section 7.

## 2  Dirac Hamiltonians, spheres and quantum geometry

We consider an important class of Bloch Hamiltonians, which are built from Dirac matrices according to

$$H_{\mathrm{D}}(\mathbf{k}) = -\sum_{i=1}^{D} d^i(\mathbf{k})\gamma_i = -\vec{d}(\mathbf{k})\cdot\vec{\gamma}\,, \tag{3}$$

where the $\gamma_i$ matrices form an irreducible representation (irrep) of the complex Clifford algebra in $D$ generators [53],

$$\gamma_i\gamma_j + \gamma_j\gamma_i = 2\delta_{ij}I\,, \quad 1 \le i,j \le D\,, \tag{4}$$

and where the momentum $\mathbf{k}$ is defined over a $d-$dimensional Brillouin zone $\mathbb{T}^d$. The Dirac matrices $\gamma_i$ have size $2^n \times 2^n$, for $n = \lfloor D/2 \rfloor$, hence leading to a multi-band spectrum $E(\mathbf{k})$. This broad class of Hamiltonians include emblematic models of topological insulators and semimetals, such as Chern insulators in two and four spatial dimensions [38,54,55], the Su-Shrieffer-Heeger (SSH) model in one dimension [36,44,56–58], chiral insulators in three dimensions [59], and Weyl semimetals in three and five dimensions [60,61]. As already emphasized in the introductory Section 1, these ideal Dirac toy models are experimentally realized in a broad class of synthetic lattice systems, including ultracold gases in optical lattices and photonics devices, where geometric and topological properties can be accessed through various techniques [58,62].

Due to the Clifford algebra relations, the spectrum of the generic Hamiltonian in Eq. (3) is directly obtained as $E(\mathbf{k}) = \pm|\vec{d}(\mathbf{k})|$; we note that the corresponding bands are degenerate in general. In the following, we consider non-interacting fermions at half-filling, which amount to setting the Fermi level at $E_F = 0$. In this framework, the spectrum crosses the Fermi level whenever the vector $\vec{d}$ vanishes. Because the vector has $D$ components, this will generically occur in a $(d-D)-$dimensional (closed) submanifold of the Brillouin zone, which we will refer to as the generalized Fermi surface $\Sigma$ [63]. For $D > d$, $\Sigma$ will, generically, be the empty set and the system will be gapped. In this work, we will consider cases where $d = 2n$ and $D = 2n + 1$, for some integer $n > 0$, corresponding to *gapped systems without chiral symmetry*, and cases where $d = 2n-1$ and $D = 2n$, corresponding to *gapped systems with chiral symmetry*. In the latter case, the chiral symmetry is implemented by the additional matrix $\gamma_{D+1} = \gamma_{2n+1}$ which, up to a multiplicative constant, is the product of all the $2n$ gamma matrices in the irrep; this matrix $\gamma_{2n+1}$ anticommutes with the Hamiltonian, hence signaling chiral symmetry. For $D = d$, $\Sigma$ consists, generically, of a finite collection of isolated points, and the system falls into the class of *Weyl-type* semimetals.

Away from $\Sigma$, one can define a unit length vector $\vec{n} = \vec{d}/|\vec{d}|$ living on a $(D-1)$-dimensional sphere $S^{D-1}$, and which completely determines the eigenstates of the Bloch Hamiltonian: at momentum $\mathbf{k}$ the eigenstates are determined by $\vec{n}(\mathbf{k}) \in S^{D-1}$. From the point of view of the eigenstates, it is sufficient to consider the Hamiltonian

$$H = -\vec{n}\cdot\vec{\gamma}\,, \ \text{with } \vec{n}\in S^{D-1}\,. \tag{5}$$

In the following, we consider the cases $D = 2n + 1$ and $D = 2n$ separately, due to the presence or absence of chiral symmetry.

Let us first consider $D = 2n + 1$. The negative energy eigenstates (i.e. the Fermi sea) are determined by the projector $P = (1/2)(I + \vec{n}\cdot\vec{\gamma})$. The quantum geometry associated with this projector, or equivalently, with the associated eigenvector spaces, is rich and it is encoded in

the *non-Abelian quantum geometric tensor* (QGT) [64], whose imaginary part is related to the non-Abelian Berry curvature [36],

$$\Omega = PdP \wedge dPP = \frac{1}{4}P\,(d\vec{n}\cdot\vec{\gamma})\wedge(d\vec{n}\cdot\vec{\gamma})P\,, \tag{6}$$

where $\wedge$ denotes the wedge product of forms (which guarantees that $\Omega$ is a completely skew-symmetric tensor). The real part of the QGT is the non-Abelian quantum metric, the trace of which reduces to the standard quantum metric [2],

$$g_{2n} = \mathrm{Tr}\,(PdPdPP) = 2^{n-3}d\vec{n}\cdot d\vec{n}\,. \tag{7}$$

Remarkably, $g_{2n}$ is the round metric of a sphere $S^{2n}$ of radius $R$ with $R^2 = 2^{n-3}$. The only non-trivial topological invariant [1] that one can associate with the projector $P$ (and in fact, with the Hamiltonian $H$) is the $n-$th Chern character, which is expressed as

$$\omega_{2n} = \frac{1}{n!}\left(\frac{i}{2\pi}\right)^n \mathrm{Tr}\,(\Omega^n)\,. \tag{8}$$

Using the properties of the $\gamma_i$ matrices [Appendix A], we obtain

$$\omega_{2n} = (-1)^n \frac{d\mathrm{vol}_{g_{2n}}}{\mathrm{vol}_{g_{2n}}(S^{2n})}\,, \tag{9}$$

where $d\mathrm{vol}_g$ is the volume form associated with the metric $g$, and $\mathrm{vol}_{g_{2n}}(S^{2n}) = \int_{S^{2n}} d\mathrm{vol}_{g_{2n}}$ is the so-called *quantum volume*. The result in Eq. (9), which connects the quantum metric $g_{2n}$ to the topological Chern character $\omega_{2n}$, is central in our work, and it will be made more explicit in the next Sections.

For $D = 2n$, chiral symmetry imposes that the Hamiltonian can be recast in the form

$$H = -\vec{n}\cdot\vec{\gamma} = \begin{bmatrix} 0 & q^{\dagger} \\ q & 0 \end{bmatrix}\,, \tag{10}$$

where $q$ is a unitary matrix. Because the irrep of the Clifford algebra in $D = 2n$ generators can be seen as the restriction of the irrep of the Clifford algebra in $2n+1$ generators, one can describe the geometry and topology of this Hamiltonian in terms of $S^{2n-1} \subset S^{2n}$, where $S^{2n-1}$ can be seen as the equator of the sphere $S^{2n}$; see Fig. 1. As a consequence, the quantum metric is exactly the restriction to the equator of Eq. (7),

$$g_{2n-1} = g_{2n}\Big|_{S^{2n-1}} = 2^{n-3}d\vec{n}\cdot d\vec{n}\,, \tag{11}$$

and it defines the round metric of a sphere $S^{2n-1}$ of radius $R$ with, as before, $R^2 = 2^{n-3}$. Here, the relevant non-trivial topological invariant that we can associate to $H$ is a winding-number class, which can be determined by dimensional reduction arguments. Specifically, it is the winding-number class of a gauge transformation on the equator $S^{2n-1}$ of $S^{2n}$, which relates eigenvectors in different gauges defined in the upper and lower hemispheres, and which is homotopic to the unitary transformation $q$. The differential form that represents the winding-number class is expressed as

$$\omega_{2n-1} = (-1)^{n-1}\left(\frac{i}{2\pi}\right)^n \frac{(n-1)!}{(2n-1)!}\mathrm{Tr}\Big[\big(q^{-1}dq\big)^{2n-1}\Big]\,. \tag{12}$$

---

[1]This statement derives from the fact that spheres only have cohomology in zero and top degrees, and the fact that the map induced by the Chern character from the complex $K$-cohomology (tensored with the rationals) to the ordinary cohomology is an isomorphism; see Ref. [65].

As in Eq. (9), the form $\omega_{2n-1}$ can be written in terms of the volume element of the quantum metric over $S^{2n-1}$,

$$\omega_{2n-1} = (-1)^n \frac{d\mathrm{vol}_{g_{2n-1}}}{\mathrm{vol}_{g_{2n-1}}(S^{2n-1})}. \tag{13}$$

The similarity with the non-chiral case can be deduced from the dimensional-reduction argument (see, for instance Ref. [66], where the variation of the Chern-Simons forms under gauge transformations, whose derivatives determine the Chern characters locally, is expressed in terms of the winding-number class of the gauge transformation), which sets $\int_{S^{2n}} \omega_{2n} = \int_{S^{2n-1}} \omega_{2n-1} = (-1)^n$.

The following Sections aim at clarifying and illustrating the results in Eqs. (9) and (13) based on relevant examples of topological matter.

## 3  Gapped systems without chiral symmetry

Here we set $d = D - 1 = 2n$ for some integer $n > 0$. In this case, $H(\mathbf{k})$ is generically gapped and there is no chiral symmetry. The vector $\vec{n}(\mathbf{k})$ is hence well-defined everywhere and it can be used to *pullback* the quantum geometry in $S^{2n}$ to the Brillouin zone $\mathbb{T}^{2n}$; see Fig. 1(a). We will write the pullback of the metric in the usual periodic coordinates $\mathbf{k} = (k_1, \ldots, k_{2n})$ defined in the Brillouin zone as

$$g = \sum_{i,j=1}^{2n} g_{ij}(\mathbf{k}) dk_i dk_j = 2^{n-3} \sum_{i,j=1}^{2n} \frac{\partial \vec{n}}{\partial k_i} \cdot \frac{\partial \vec{n}}{\partial k_j} dk_i dk_j. \tag{14}$$

The non-trivial topological invariant associated with $H(\mathbf{k})$ is given by pulling back the topological invariant over $S^{2n}$. Now because $D - 1 = d$, the pullback of the volume form $d\mathrm{vol}_{g_{2n}}$ is, up to the sign of the Jacobian of the transformation at each point, the volume form of the pullback metric $d\mathrm{vol}_g = \sqrt{\det(g)} dk_1 \wedge \cdots \wedge dk_{2n}$. Furthermore, because the Berry curvature in momentum space is the pullback of the Berry curvature in $S^{2n}$, it follows that the pullback of $\omega_{2n}$ is the $n$—th Chern character of the occupied Bloch bundle. In local coordinates, and using $\Omega = \frac{1}{2} \sum_{i,j=1}^{2n} \Omega_{ij} dk_i \wedge dk_j$, the identity in Eq. (9) now takes the more explicit form

$$\frac{i^n}{(2\pi)^n n!} \frac{1}{2^n} \sum_{i_1, j_1, \ldots, i_n, j_n = 1}^{2n} \mathrm{Tr}\left(\Omega_{i_1 j_1} \ldots \Omega_{i_n j_n}\right) \varepsilon^{i_1 j_1 \ldots i_n j_n} = \mathrm{sgn}(d\vec{n})(-1)^n \frac{(2n)!}{2^{n(n-1)+1} n! \pi^n} \sqrt{\det(g)}, \tag{15}$$

where $\mathrm{sgn}(d\vec{n}) = \pm 1$ depending on whether the map to the sphere induced by $\vec{n}$ is orientation preserving or reversing at the considered point of the Brillouin zone, and we have used $\mathrm{vol}_{g_{2n}}(S^{2n}) = (2^{n(n-1)+1} \pi^n n!)/(2n)!$. We note that $\mathrm{sgn}(d\vec{n})$ can be explicitly computed as

$$\mathrm{sgn}(d\vec{n}) = \mathrm{sgn}\left( \sum_{i_1 \ldots i_{2n+1} = 1}^{2n+1} \varepsilon_{i_1 \ldots i_{2n+1}} d^{i_1} \frac{\partial d^{i_2}}{\partial k_1} \cdots \frac{\partial d^{i_{2n+1}}}{\partial k_{2n}} \right). \tag{16}$$

Eq. (15) is one of the main results of this work, relating the Chern character to the determinant of the quantum metric. We note that similar relations between the Berry curvature and the geometry of the sphere were recently reported in Refs. [67–69].

Importantly, the left-hand side of Eq. (15) integrates to an integer topological invariant of the occupied Bloch band, which completely classifies the topological phase,

$$\mathrm{Ch}_n = \frac{1}{n!} \left( \frac{i}{2\pi} \right)^n \int_{\mathbb{T}^{2n}} \mathrm{Tr}(\Omega^n) \in \mathbb{Z}, \tag{17}$$

and which is known as the $n$-th Chern number. The equality established in Eq. (15) implies the inequality

$$|\text{Ch}_n| \leq \frac{(2n)!}{2^{n(n-1)+1}n!\pi^n}\text{vol}_g(\mathbb{T}^{2n}),\tag{18}$$

where we have identified the *quantum volume* $\text{vol}_g(\mathbb{T}^{2n}) = \int_{\mathbb{T}^{2n}}\sqrt{\det(g)}d^{2n}k$; the latter quantity is also known as the *complexity* of the band [22]. We point out that the equality is satisfied provided $\text{sgn}(d\vec{n})$ is constant (everywhere where it is meaningful, i.e., where $\sqrt{\det(g)} \neq 0$), or, equivalently, if the function on the left-hand side of Eq. (15) does not change sign. Note also that $\sqrt{\det(g)}d^{2n}k$ is not a volume form in the strict mathematical sense because it necessarily vanishes somewhere in the Brillouin zone. The reason is that $\vec{n}: \mathbb{T}^{2n} \to S^{2n}$ cannot be an immersion since the fundamental group of the torus is non-trivial while that of the sphere is trivial; see the proof of Theorem 3 of Ref. [41] for the case $n=1$ which is readily generalized to this case. From Eq. (18), one immediately concludes that if the quantum volume is smaller than 1, then the system is certainly adiabatically connected to the trivial insulator.

In the following, we illustrate the significance of this result by considering paradigmatic examples of Chern insulators in $d = 2$ and $d = 4$ dimensions.

## 3.1 Chern insulators in $d = 2$ dimensions

A generic 2-band Chern insulator model in two dimensions reads

$$H(\mathbf{k}) = \epsilon(\mathbf{k})I + d_x(\mathbf{k})\sigma^1 + d_y(\mathbf{k})\sigma^2 + d_z(\mathbf{k})\sigma^3,\tag{19}$$

where the matrices $\sigma^{1,2,3}$ are Pauli matrices; see Refs. [54, 55] and below for an explicit example. The quantum metric identifies a 2-sphere of radius $1/2$ [Eq. (7)], and its relation to the Chern character (i.e. the Berry curvature in the lower band) simply reads [Eq. (15)]

$$\frac{i}{2\pi}\Omega_{12} = \text{sgn}(d\vec{n})\frac{\sqrt{\det(g)}}{\pi},\tag{20}$$

where we used the fact that the antipodal map $\vec{n} \mapsto -\vec{n}$ is orientation reversing. In this particular case, note that $\text{sgn}(d\vec{n})$ can be recast in the familiar triple product formula

$$\text{sgn}(d\vec{n}) = \text{sgn}\left[\vec{d} \cdot \left(\frac{\partial\vec{d}}{\partial k_1} \times \frac{\partial\vec{d}}{\partial k_2}\right)\right].\tag{21}$$

Here, Eq. (18) reduces to

$$|\text{Ch}_1| \leq \frac{\text{vol}_g(\mathbb{T}^2)}{\pi},\tag{22}$$

as was previously studied in Refs. [41, 42].

**An explicit Chern insulator in $d = 2$ dimensions**

Here we consider a massive Dirac model in two dimensions as a representative of a Chern insulator. The Bloch Hamiltonian reads [38, 55]

$$H(\mathbf{k}) = \sum_{i=1}^{2}\sin(k_i)\sigma^i + \left(M - \sum_{i=1}^{2}\cos(k_i)\right)\sigma^3 = \vec{d}(\mathbf{k}) \cdot \vec{\sigma},\tag{23}$$

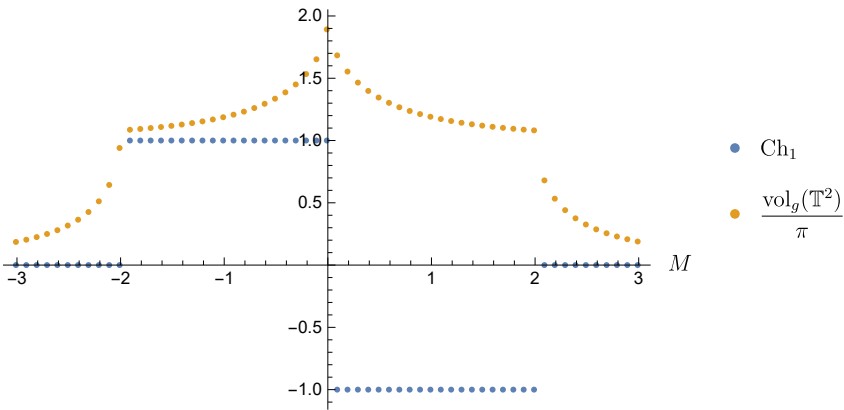

Figure 2: Chern number and quantum volume as a function of $M$.

where $M$ is the mass parameter. The spectrum of the Hamiltonian is given by $E(\mathbf{k}) = \pm|\vec{d}|$, where $|\vec{d}| = \sqrt{\sin^2(k_1) + \sin^2(k_2) + (M - \cos(k_1) - \cos(k_2))^2}$. We now explicitly verify the relation in Eq. (20) using the quantum metric

$$g = \sum_{i,j=1}^{2} g_{ij}(\mathbf{k}) dk_i dk_j = \frac{1}{4} \sum_{i,j=1}^{2} \frac{\partial \vec{n}}{\partial k_i} \cdot \frac{\partial \vec{n}}{\partial k_j} dk_i dk_j, \tag{24}$$

where $\vec{n} = \vec{d}/|\vec{d}|$. The right-hand side of Eq. (20) involves

$$\frac{\sqrt{\det(g)}}{\pi} = \frac{1}{4\pi} \frac{|(\cos(k_1) + \cos(k_2) - M \cos(k_1) \cos(k_2))|}{\left(\sin^2(k_1) + \sin^2(k_2) + (M - \cos(k_1) - \cos(k_2))\right)^{3/2}}, \tag{25}$$

and

$$\text{sgn}(d\vec{n}) = \text{sgn}\left[\vec{d} \cdot \left(\frac{\partial \vec{d}}{\partial k_1} \times \frac{\partial \vec{d}}{\partial k_2}\right)\right] = \text{sgn}\left[-(\cos(k_1) + \cos(k_2) - M \cos(k_1) \cos(k_2))\right]. \tag{26}$$

Altogether, we find

$$\text{sgn}(d\vec{n}) \frac{\sqrt{\det(g)}}{\pi} = -\frac{1}{4\pi} \frac{\cos(k_1) + \cos(k_2) - M \cos(k_1) \cos(k_2)}{\left(\sin^2(k_1) + \sin^2(k_2) + (M - \cos(k_1) - \cos(k_2))\right)^{3/2}}. \tag{27}$$

Besides, the left-hand side of Eq. (20) reads

$$\frac{i\Omega_{12}}{2\pi} = \frac{1}{4\pi} \frac{\vec{d} \cdot \left(\frac{\partial \vec{d}}{\partial k_1} \times \frac{\partial \vec{d}}{\partial k_2}\right)}{|d|^3} = -\frac{1}{4\pi} \frac{\cos(k_1) + \cos(k_2) - M \cos(k_1) \cos(k_2)}{\left(\sin^2(k_1) + \sin^2(k_2) + (M - \cos(k_1) - \cos(k_2))\right)^{3/2}}, \tag{28}$$

which coincides with Eq. (27), in agreement with the relation in Eq. (20).

The topological phase diagram of this model is captured by the first Chern number: $\text{Ch}_1 = 0$ for $|M| > 2$, $\text{Ch}_1 = 1$ for $-2 < M < 0$ and $\text{Ch}_1 = -1$ for $0 < M < 2$. The inequality involving the Chern number and the quantum volume, Eq. (22), is illustrated in Fig. 2.

These results are consistent with the findings of Ref. [42], where more instances of $d = 2$ insulators are presented.

## 3.2 Chern insulators in $d = 4$ dimensions

Let us now apply our formula to a higher-dimensional Chern insulator, by considering the following 4-band model in four dimensions [38]

$$H(\mathbf{k}) = d_0(\mathbf{k})\Gamma^0 + d_1(\mathbf{k})\Gamma^1 + d_2(\mathbf{k})\Gamma^2 + d_3(\mathbf{k})\Gamma^3 + d_4(\mathbf{k})\Gamma^4, \tag{29}$$

where the five matrices $\Gamma^{0,1,2,3,4}$ are $4 \times 4$ Dirac matrices and where the momenta span a four-dimensional Brillouin zone. In this case, the quantum metric identifies a 4-sphere of radius $\frac{1}{\sqrt{2}}$ [Eq. (7)], and the relevant topological invariant is the *second* Chern number associated with the doubly-degenerate low-energy band [38]. We recall that this invariant, which was measured in cold atoms [70, 71], plays a central role in the 4D quantum Hall effect [72, 73]. According to our formula, the corresponding 2nd Chern character is directly related to the quantum metric according to the relation [Eq. (15)]

$$-\frac{1}{32\pi^2} \sum_{i,j,k,l=1}^{4} \mathrm{Tr}\left(\Omega_{ij}\Omega_{kl}\right)\varepsilon^{ijkl} = -\mathrm{sgn}(d\vec{n})\frac{3\sqrt{\det(g)}}{2\pi^2}. \tag{30}$$

This result was independently obtained in Ref. [68]. Here, Eq. (18) yields an inequality between the second Chern number and the quantum volume

$$|\mathrm{Ch}_2| \leq \frac{3\mathrm{vol}_g(\mathbb{T}^4)}{2\pi^2}. \tag{31}$$

**A time-reversal-invariant insulator in $d = 4$ dimensions**

We consider the following Bloch Hamiltonian representing a 4−dimensional Chern insulator with time-reversal symmetry [38]

$$H(\mathbf{k}) = \sum_{i=1}^{4} \sin(k_i)\Gamma^i + \left(M - \sum_{i} \cos(k_i)\right)\Gamma^5 = \vec{d}(\mathbf{k}) \cdot \vec{\Gamma}. \tag{32}$$

The phase diagram of this model is captured by the second Chern number: $\mathrm{Ch}_2 = 0$ for $|M| > 4$, $\mathrm{Ch}_2 = +1$ for $-4 < M < -2$, $\mathrm{Ch}_2 = -3$ for $-2 < M < 0$, $\mathrm{Ch}_2 = +3$ for $0 < M < 2$, and $\mathrm{Ch}_2 = -1$ for $2 < M < 4$. The inequality involving the second Chern number and the quantum volume, Eq. (31), is illustrated in Fig. 3.

## 4 Gapped systems with chiral symmetry

We now set $d = D - 1 = 2n - 1$ for some integer $n > 0$. In this case, $H(\mathbf{k})$ is again generically gapped but chiral symmetry is now present. As above, the vector $\vec{n}(\mathbf{k})$ is well-defined everywhere and it can be used to pullback the quantum geometry in $S^{2n-1}$ to the Brillouin torus $\mathbb{T}^{2n-1}$. The pullback of the metric in the periodic coordinates $\mathbf{k} = (k_1, \ldots, k_{2n-1})$ is written as above,

$$g = \sum_{i,j=1}^{2n-1} g_{ij}(\mathbf{k})dk_i dk_j = 2^{n-3} \sum_{i,j=1}^{2n-1} \frac{\partial \vec{n}}{\partial k_i} \cdot \frac{\partial \vec{n}}{\partial k_j} dk_i dk_j. \tag{33}$$

The non-trivial topological invariant associated with $H(\mathbf{k})$ is given by pulling back the topological invariant represented by $\omega_{2n-1}$ over $S^{2n-1}$. Again, since $D - 1 = d$, the pullback of the volume form $d\mathrm{vol}_{g_{2n}}$ is, up to the sign of the Jacobian of the transformation at each point, the

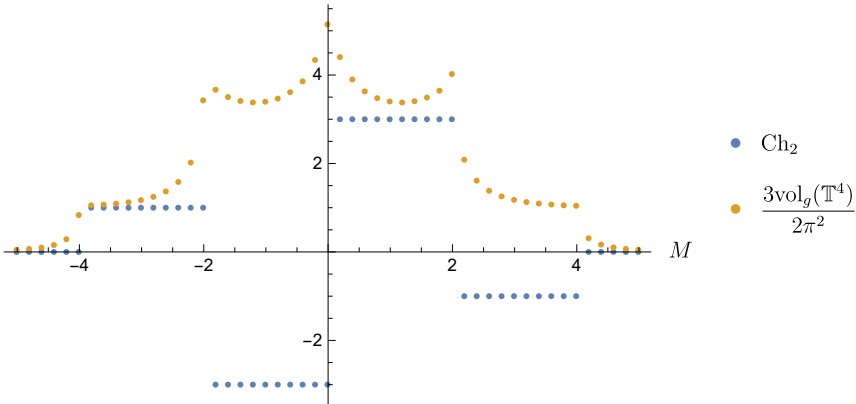

Figure 3: Second Chern number and quantum volume as a function of the model parameter $M$.

volume form of the pullback metric $d\text{vol}_g = \sqrt{\det(g)}dk_1 \wedge \cdots \wedge dk_{2n}$. This leads to our second main result,

$$
\begin{aligned}
&(-1)^{n-1}\left(\frac{i}{2\pi}\right)^n \frac{(n-1)!}{(2n-1)!}\sum_{i_1\ldots i_{2n-1}=1}^{2n-1}\text{Tr}\left(q^{-1}\frac{\partial q}{\partial k_{i_1}}\cdots q^{-1}\frac{\partial q}{\partial k_{i_{2n-1}}}\right)\varepsilon^{i_1\ldots i_{2n-1}}\\
&= \text{sgn}(d\vec{n})(-1)^n\frac{(n-1)!}{2^{\frac{1}{2}(n-1)(2n-5)}\pi^n}\sqrt{\det(g)},
\end{aligned}
\tag{34}
$$

which connects the winding-number class to the quantum metric. Here, $\text{sgn}(d\vec{n})=\pm 1$ depending on whether the map to the sphere induced by $\vec{n}$ is orientation preserving or reversing at the considered point of the Brillouin zone, and we have used $\text{vol}_{g_{2n-1}}(S^{2n-1}) = 2^{\frac{1}{2}(n-1)(2n-5)}/(n-1)!$. The quantity $\text{sgn}(d\vec{n})$ can now be computed as

$$
\text{sgn}(d\vec{n}) = \text{sgn}\left(\sum_{i_1\ldots i_{2n}=1}^{2n}\varepsilon_{i_1\ldots i_{2n}}d^{i_1}\frac{\partial d^{i_2}}{\partial k_1}\cdots\frac{\partial d^{i_{2n}}}{\partial k_{2n-1}}\right).
\tag{35}
$$

The left-hand side of Eq. (34) integrates to an integer topological invariant of the occupied Bloch band, the winding number $\nu$ of the map $q$ [Eq. (10)], which completely classifies the topological phase,

$$
\nu = (-1)^{n-1}\left(\frac{i}{2\pi}\right)^n\frac{(n-1)!}{(2n-1)!}\int_{\mathbb{T}^{2n}}\text{Tr}\left[\left(q^{-1}dq\right)^{2n-1}\right]\in\mathbb{Z}.
\tag{36}
$$

The equality established in Eq. (34) now implies an inequality for the winding number

$$
|\nu| \le \frac{(n-1)!}{2^{\frac{1}{2}(n-1)(2n-5)}\pi^n}\text{vol}_g(\mathbb{T}^{2n-1}).
\tag{37}
$$

Again, we note that the equality is satisfied whenever $\text{sgn}(d\vec{n})$ is constant (for $\sqrt{\det(g)}\neq 0$), or, equivalently, if the function on the left-hand side of Eq. (34) does not change sign. We will now illustrate these results with prime examples of chiral insulators.

## 4.1 Chiral insulators in $d=1$ dimension

The simplest instance of a chiral insulator is provided by the generic Hamiltonian

$$
H(k) = d_x(k)\sigma^1 + d_y(k)\sigma^2,
\tag{38}
$$

which describes the emblematic Su-Shrieffer-Heeger (SSH) model in one dimension [36,56]; see below. The $1d$ chiral insulator is characterized by a quantized Zak phase [44,57,58], which defines a topological winding number in the 1D Brillouin zone. Using the formula in Eq. (34) and using the fact that in this case the antipodal map is orientation preserving, we find that the corresponding winding-number class is related to the quantum metric in the lower band according to

$$\frac{i}{2\pi} q^{-1} \frac{\partial q}{\partial k} = -\text{sgn}(d\vec{n}) \frac{\sqrt{g_{11}}}{\pi}, \tag{39}$$

where $q = (d_x + i d_y)/|d_x + i d_y|$, and

$$\text{sgn}(d\vec{n}) = \text{sgn}\left( d_x \frac{\partial d_y}{\partial k} - d_y \frac{\partial d_x}{\partial k} \right). \tag{40}$$

In this case, the inequality involving the winding number in Eq. (37) takes the simple form

$$|v| \leq \frac{\text{vol}_g(\mathbb{T}^1)}{\pi}. \tag{41}$$

**The Su-Schrieffer-Heeger (SSH) model**

We now address the SSH model in detail to illustrate this $d = 1$ case. The Bloch Hamiltonian in Eq. (38) is specified by the vector

$$\vec{d}(k) = (v + w\cos(k), w\sin(k)), \tag{42}$$

where $v, w$ are the two alternating hopping amplitudes in the SSH lattice [58]. The relative strength of the hoppings $|v/w|$ determines the topological character of the model. The spectrum of the model is given by $E(k) = \pm|\vec{d}(k)| = \pm\sqrt{v^2 + w^2 + 2vw\cos(k)}$, the latter being gapless when $|v/w| = 1$. The quantum metric is readily evaluated and reads

$$g = \frac{1}{4} \frac{w^2 (w + v\cos(k))^2}{(v^2 + w^2 + 2vw\cos(k))^2} dk^2. \tag{43}$$

We now illustrate the relation in Eq. (39), connecting the winding-number class to the quantum metric. For the SSH model, the right-hand side of Eq. (39) reads

$$-\text{sgn}(d\vec{n}) \frac{\sqrt{g_{11}}}{\pi} = -\text{sgn}(d\vec{n}) \frac{1}{2\pi} \frac{|w(w + v\cos(k))|}{v^2 + w^2 + 2vw\cos(k)}, \tag{44}$$

where

$$\text{sgn}(d\vec{n}) = \text{sgn}\left( d_x \frac{\partial d_y}{\partial k} - d_y \frac{\partial d_x}{\partial k} \right)$$
$$= \text{sgn}\left[ w(w + v\cos(k)) \right]. \tag{45}$$

Altogether, this yields

$$-\text{sgn}(d\vec{n}) \frac{\sqrt{g_{11}}}{\pi} = -\frac{1}{2\pi} \frac{w(w + v\cos(k))}{v^2 + w^2 + 2vw\cos(k)}. \tag{46}$$

Besides, the left-hand side of Eq. (39) can be obtained from $q = \frac{d_x + i d_y}{|d_x + i d_y|}$, yielding

$$\frac{i}{2\pi} q^{-1} \frac{\partial q}{\partial k} = -\frac{1}{2\pi} \frac{d_x \frac{\partial d_y}{\partial k} - d_y \frac{\partial d_x}{\partial k}}{d_x^2 + d_y^2} = -\frac{1}{2\pi} \frac{w(w + v\cos(k))}{v^2 + w^2 + 2vw\cos(k)}, \tag{47}$$

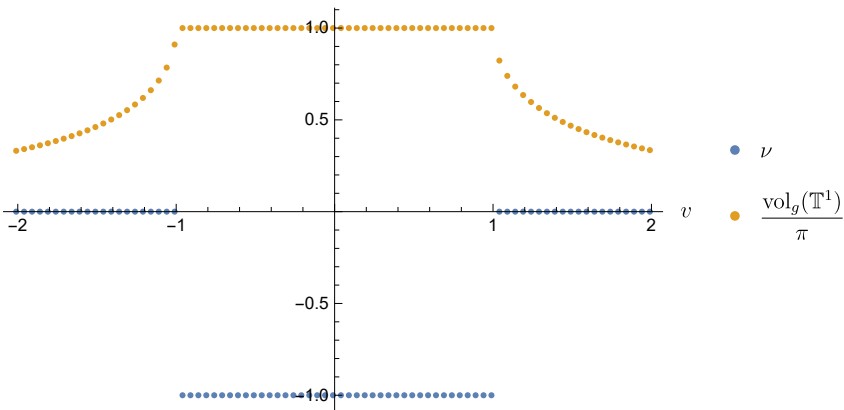

Figure 4: Winding number and quantum volume as a function of $v$, with $w = 1$.

in agreement with Eq. (46) and the relation in Eq. (39).

Without loss of generality, we may take $w = 1$, in which case the phase diagram is dictated by the winding number as follows: $\nu = 0$ for $|v| > 1$ and $\nu = -1$ for $|v| < 1$. For $\nu = 0$, which is located in the non-trivial regime, the quantum metric assumes the simple form

$$g_{11} = \frac{1}{4}dk^2, \tag{48}$$

and the winding-number class is simply represented by

$$\frac{i}{2\pi}q^{-1}dq = -\frac{\sqrt{g_{11}}dk}{\pi} = -\frac{dk}{2\pi}. \tag{49}$$

We illustrate the inequality involving the winding number and the quantum volume [Eq. (41)] in Fig. 4. We note that the right-hand side of this inequality is smaller than 1 for $|v| > 1$, hence implying $\nu = 0$, and it is exactly equal to 1 for $|v| < 1$; this is consistent with the fact that $\text{sgn}(d\vec{n})$ is constant as a function of $k$ in that region.

## 4.2 Chiral insulators in $d = 3$ dimensions

Moving on to higher-dimensions, a model describing a chiral insulator in $d = 3$ dimensions can be written in the form

$$H(\mathbf{k}) = d_0(\mathbf{k})\Gamma^0 + d_1(\mathbf{k})\Gamma^1 + d_2(\mathbf{k})\Gamma^2 + d_3(\mathbf{k})\Gamma^3, \tag{50}$$

where the $\Gamma$'s represent the $4 \times 4$ Dirac matrices; see Ref. [59] for an explicit model and implementation. Here, the quantum metric identifies a 3-sphere, and the relevant topological invariant characterizing the insulator is provided by the 3D winding number [74]. Using the formula in Eq. (34), the relation between this higher-dimensional winding-number class and the quantum metric reads

$$\frac{1}{24\pi^2}\sum_{i,j,k=1}^{3}\text{Tr}\left(q^{-1}\frac{\partial q}{\partial k_i}q^{-1}\frac{\partial q}{\partial k_j}q^{-1}\frac{\partial q}{\partial k_k}\right)\varepsilon^{ijk} = \text{sgn}(d\vec{n})\frac{\sqrt{2}\sqrt{\det(g)}}{\pi^2}, \tag{51}$$

where we used the fact that the antipodal map is orientation preserving between spheres of odd dimensions. The inequality in Eq. (37) reduces to

$$|\nu| \leq \frac{\sqrt{2}\text{vol}_g(\mathbb{T}^3)}{\pi^2}, \tag{52}$$

where $\nu$ is the winding number in $d = 3$ dimensions.

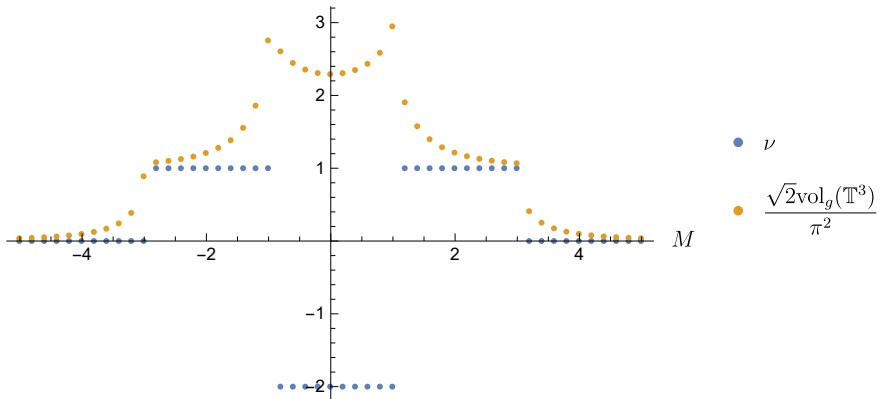

Figure 5: $3d$ winding number and quantum volume as a function of $M$.

**An explicit chiral insulator in $d = 3$ dimensions**

We consider the following Bloch Hamiltonian, describing the chiral insulator of Ref. [59],

$$H(\mathbf{k}) = \sum_{i=1}^{3} t_{\text{SO}} \sin(k_i) \Gamma^i + (m_z - t_0 (\cos(k_1) + \cos(k_2) + \cos(k_3))) \Gamma^0, \tag{53}$$

where the physical meaning of the parameters $t_{\text{SO}}$, $m_z$ and $t_0$ is not relevant for this discussion. For the sake of presentation, we hereby set $t_{\text{SO}} = t_0 = 1$, and $M \equiv m_z$, so that the above equation reduces to

$$H(\mathbf{k}) = \sum_{i=1}^{3} \sin(k_i) \gamma_i + (M - (\cos(k_1) + \cos(k_2) + \cos(k_3))) \gamma_0 = \vec{d}(\mathbf{k}) \cdot \vec{\gamma}, \tag{54}$$

which is a higher dimensional analogue of the massive Dirac model considered in Section 3.1. The phase diagram of this model is established by the 3D winding number: $\nu = 0$ for $|M| > 3$, $\nu = +1$ for $-3 < M < -1$, $\nu = -2$ for $-1 < M < 1$, and $\nu = +1$ for $1 < M < 3$. We illustrate the inequality involving the 3D winding number and the quantum volume [Eq. (52)] in Fig. 5.

## 5 Gapless Weyl-type systems

A prime example of a gapless Weyl-type system is provided by the 3D Weyl Hamiltonian [60]

$$H(\mathbf{k}) = k_x \sigma^1 + k_y \sigma^2 + k_z \sigma^3. \tag{55}$$

The Berry curvature field in the lower band emanates radially from the origin $\mathbf{k} = 0$, and its flux through a 2-sphere $S^2$ that contains the origin is quantized according to the first Chern number

$$\text{Ch}_1 = \int_{S^2} \frac{i}{2\pi} \Omega = 1. \tag{56}$$

As pointed out in Ref. [24], the quantum metric in the lower band identifies a 2-sphere and is related to the Berry curvature (using positively oriented coordinates on the sphere) as

$$i\Omega_{ij} = 2\sqrt{\det(g)} \varepsilon_{ij}, \quad 1 \leq i, j \leq 2. \tag{57}$$

This is the simplest form of Eqs. (9) and (15), where the Brillouin zone is now replaced by a sphere surrounding the monopole. Accordingly, the topological "monopole" charge can be expressed in terms of the quantum metric as

$$\text{Ch}_1 = \frac{\text{vol}_{g_2}(S^2)}{\pi} = 1.$$  (58)

Weyl-type systems can also be generalized to higher dimensions [61, 75]. For instance, a generic 5D Weyl Hamiltonian can be written in terms of the $4 \times 4$ Dirac matrices as [61]

$$H(\mathbf{k}) = k_0 \Gamma^0 + k_1 \Gamma^1 + k_2 \Gamma^2 + k_3 \Gamma^3 + k_4 \Gamma^4.$$  (59)

In this case, the origin hosts a Yang (non-Abelian) monopole whose topological charge is provided by the second Chern number. Using the formula in Eq. (15), we find that this monopole charge can be obtained from the quantum metric as

$$\text{Ch}_2 = \frac{1}{2}\left(\frac{i}{2\pi}\right)^2 \int_{S^4} \text{Tr}\left(\Omega^2\right) = -\frac{3}{2\pi^2} \text{vol}_{g_4}(S^4) = -1,$$  (60)

where the Brillouin zone $\mathbb{T}^4$ was replaced by a sphere $S^4$. The quantum metric identifies a 4-sphere of radius $\frac{1}{\sqrt{2}}$ that surrounds the Yang monopole, whose topological charge density is a multiple of the quantum volume form.

# 6  Relation to the Cramér-Rao bound and measurement uncertainty

In quantum metrology, the quantum Fisher information of a pure state, defined over a parameter space, is equivalent to the quantum metric [2–5]. In that metrological context, the quantum metric thus plays a key role by constraining the precision of quantum-parameter-estimation measurements through the celebrated Cramér-Rao bound [51, 52]. For a one-dimensional space spanned by the parameter $\beta$, this relation reads

$$\delta\beta \geq 1/\sqrt{N\mathcal{F}_\beta},$$  (61)

where $\delta\beta$ denotes the uncertainty associated with the parameter-estimation measurement, $\mathcal{F}_\beta = 4g_{\beta\beta}$ is the quantum Fisher information ($g_{\beta\beta}$ denotes the quantum metric), and $N$ is the number of independent measurements.

The present work introduced relations between the quantum metric and topological classes, and it is thus intriguing to explore how the latter can influence metrological properties. In this Dirac-Hamiltonian framework, the relevant parameter space is provided by the momentum space [Section 2], over which one defines the quantum metric and the quantum-parameter-estimation measurement.

In this Section, we address this question by considering the case of gapped systems without chiral symmetry of dimension $d = 2n$; see Section 3. The relevant states for the parameter-estimation measurements are defined at momentum $\mathbf{k} \in \mathbb{T}^d$ in the lowest energy band of the Dirac Hamiltonian, as described by the vector $\vec{d}(\mathbf{k})$. The Cramér-Rao bound, see Theorem 3.1 of Ref. [52], then states that the uncertainty, as measured by the covariance matrix of an unbiased estimator $\Sigma(\mathbf{k}) = [\langle \delta k_i \delta k_j \rangle]_{1 \leq i,j \leq d}$, is related to the quantum Fisher information matrix or, equivalently, to the quantum metric $g(\mathbf{k}) = [g_{ij}(\mathbf{k})]_{1 \leq i,j \leq d}$, through the inequality

$$\Sigma(\mathbf{k}) \geq \frac{1}{4N} g^{-1}(\mathbf{k}),$$  (62)

where $g^{-1}(\mathbf{k})$ denotes the inverse matrix of $g(\mathbf{k})$, $N$ is the number of independent measurements, and the inequality is understood as an inequality between positive definite matrices. The inequality holds away from the set of points in the Brillouin zone where the metric is not invertible [i.e., where the differential of the map $\vec{n}: \mathbb{T}^d \to S^d$ is not invertible].

We proceed to prove a relation between $\Sigma$ and the Chern character class representative. Using the fact that for non-negative real $d \times d$ symmetric matrices $A, B$ we have the Minkowski determinant theorem (Theorem 4.1.8 of Ref. [76]), $\det(A + B)^{1/d} \geq \det(A)^{1/d} + \det(B)^{1/d}$ it follows that $\det(A + B) \geq \det(A) + \det(B)$. Now take $A = \Sigma(\mathbf{k}) - \frac{1}{4N} g^{-1}(\mathbf{k}) \geq 0$ and $B = \frac{1}{4N} g^{-1}(\mathbf{k})$. It follows that

$$\det(A + B) = \det(\Sigma(\mathbf{k})) \geq \det\left(\Sigma(\mathbf{k}) - \frac{1}{4N} g^{-1}(\mathbf{k})\right) + \det\left(\frac{1}{4N} g^{-1}(\mathbf{k})\right)$$
$$\geq \det\left(\frac{1}{4N} g^{-1}(\mathbf{k})\right) = \frac{1}{2^{2d} N^d} \frac{1}{\det(g(\mathbf{k}))}.$$

Taking square roots, we find the inequality

$$\sqrt{\det(\Sigma(\mathbf{k}))} \geq \frac{1}{2^d \sqrt{N^d}} \frac{1}{\sqrt{\det(g(\mathbf{k}))}}. \tag{63}$$

We remark that the Cramér-Rao inequality in Eq. (62), and hence in Eq. (63), is saturated if and only if the Berry curvature is trivial, $\Omega(\mathbf{k}) = [\Omega_{ij}(\mathbf{k})]_{1 \leq i,j \leq d} = 0$; see Theorem 3.2 and Corollary 3.2.1. in Ref. [52]. It follows that, for the case at hand, due to the strict relation between the quantum metric and Berry curvature provided in Eq. (15), it is not possible to saturate the Cramér-Rao bound—hence the inequalities are strict in this case and we will drop the equality sign from hereon.

Taking absolute values on both sides of Eq. (15), we get

$$\frac{1}{(2\pi)^n n!} \frac{1}{2^n} \left| \sum_{i_1,j_1,\ldots,i_n,j_n=1}^{2n} \mathrm{Tr}\left(\Omega_{i_1 j_1} \ldots \Omega_{i_n j_n}\right) \varepsilon^{i_1 j_1 \ldots i_n j_n} \right| = \frac{(2n)!}{2^{n(n-1)+1} n! \pi^n} \sqrt{\det(g)},$$

or equivalently

$$\sqrt{\det(g)} = \frac{2^{n^2-3n+1}}{(2n)!} \left| \sum_{i_1,j_1,\ldots,i_n,j_n=1}^{2n} \mathrm{Tr}\left(\Omega_{i_1 j_1} \ldots \Omega_{i_n j_n}\right) \varepsilon^{i_1 j_1 \ldots i_n j_n} \right|. \tag{64}$$

Plugging this into the inequality of Eq. (63), we find

$$\sqrt{\det(\Sigma(\mathbf{k}))} > \frac{(2n)!}{N^n 2^{n^2-n+1}} \frac{1}{\left| \sum_{i_1,j_1,\ldots,i_n,j_n=1}^{2n} \mathrm{Tr}\left(\Omega_{i_1 j_1} \ldots \Omega_{i_n j_n}\right) \varepsilon^{i_1 j_1 \ldots i_n j_n} \right|}. \tag{65}$$

Consequently, we find that the Chern character class imposes a lower bound to the uncertainty volume at $\mathbf{k}$, as described by $\sqrt{\det(\Sigma(\mathbf{k}))}$. To show the significance of this result, we consider the two-dimensional case ($n = 1$), for which the above inequality reduces to

$$\sqrt{\det(\Sigma(\mathbf{k}))} > \frac{1}{2N} \frac{1}{|\Omega_{12}(\mathbf{k})|}. \tag{66}$$

If we recall that the Berry curvature acts like an effective magnetic field in k-space, the above relation (66) can be viewed as the constraint imposed by the corresponding effective (and local) magnetic length onto the momentum-estimation measurement. This relation is to be compared with the characteristic width (and hence, the uncertainty area) of Landau levels in real space, which is established by the magnetic length, $\delta r \sim l_B \sim 1/\sqrt{B}$.

# 7 Concluding remarks

The general relations derived in this work indicate that the topological indices characterizing topological insulators and semimetals are directly connected to the underlying quantum metric, within the framework of Dirac Hamiltonians. While this class of systems was originally introduced as toy models in condensed matter physics, they are today realized in a broad class of synthetic systems, including ultracold gases in optical lattices, solid state qubits and photonics devices. Importantly, the quantum metric can be extracted in these synthetic systems, for instance, by monitoring excitation rates upon periodic modulations [23, 77]; this is formally equivalent to measuring dynamical susceptibilities [77, 78]. These measurements were recently performed in various quantum-engineered settings [28, 29, 31, 33, 34]; see Ref. [79] for a generalization of this probing method to the case of degenerate (non-Abelian) systems, which is indeed relevant for extracting the quantum metric of higher-dimensional Dirac systems discussed in this work. Altogether, this indicates that a wide range of topological indices (including winding numbers) could be accessed in quantum-engineered systems through quantum-metric (dynamical susceptibility) measurements, including higher-dimensional settings [48, 62, 73, 80]. In this context, an interesting perspective concerns the extension of our framework to other classes of Hamiltonians, such as those based on Gell-Mann matrices [24, 74, 81–83], where similar relations between topology and quantum geometry have been identified [24]. Another interesting route concerns the implications of metric-curvature relations for quantum metrology, as we briefly illustrated in the previous Section 6.

## Acknowledgements

The authors acknowledge discussions with P. Hauke, M. Kolodrubetz, T. Ozawa and G. Palumbo. N.G. is supported by the FRS-FNRS (Belgium) and the ERC Starting Grant TopoCold. B.M. acknowledges the support from SQIG – Security and Quantum Information Group, the Instituto de Telecomunicações (IT) Research Unit, Ref. UIDB/50008/2020, funded by Fundação para a Ciência e a Tecnologia (FCT), European funds, namely, H2020 project SPARTA, as well as projects QuantMining POCI-01-0145-FEDER-031826 and PREDICT PTDC/CCI-CIF/29877/2017.

## A Computation of the $n-$th Chern character

We consider

$$H(\mathbf{k}) = -\sum_{i=1}^{2n+1} d^i(\mathbf{k})\gamma_i = -\vec{d}(\mathbf{k})\cdot\vec{\gamma}, \tag{67}$$

where the $\gamma$ matrices form an irreducible representation of the Clifford algebra in $2n+1$ generators. The Berry curvature of the negative energy bundle, described by the orthogonal projector $P = (1/2)(I + \vec{n}\cdot\vec{\gamma})$, is given by the explicit formula

$$\Omega = PdP \wedge dPP = \frac{1}{4}P(d\vec{n}\cdot\vec{\gamma}) \wedge (d\vec{n}\cdot\vec{\gamma})P = \frac{1}{4}\sum_{i,j=1}^{2n+1} P\gamma_i\gamma_j P dn^i \wedge dn^j. \tag{68}$$

The group $SO(2n+1)$ acts on $S^{2n}$ transitively. Let $R = [R_j^i]_{1 \le i,j \le 2n+1} \in SO(2n+1)$ and consider the associated map $R : S^{2n} \to S^{2n}$. Then the pullback of the Berry curvature under $R$ is

$$
\begin{aligned}
R^*\Omega &= \frac{1}{4} P \circ R (dR\vec{n} \cdot \vec{\gamma}) \wedge (dR\vec{n} \cdot \vec{\gamma}) P \circ R \\
&= U^{-1}\left(P\gamma_i\gamma_j P\right) U dn^i \wedge dn^j \\
&= U^{-1}\Omega U ,
\end{aligned}
\tag{69}
$$

where $P \circ R = \frac{1}{2}(I + (R\vec{n}) \cdot \vec{\gamma})$ and $U\gamma_i U^{-1} = R \cdot \gamma_i = \sum_{j=1}^{2n+1} R_i^j \gamma_j$, $i = 1,\dots,2n+1$, for any of the two choices of $U$ lifting $R$ to $\mathrm{Spin}(2n+1)$. As a consequence, we realize that $\Omega$ is rotation covariant in the sense that the pullback by rotations can be realized by acting by conjugation by an appropriate element of $\mathrm{Spin}(2n+1)$.

We are left with the computation of

$$
\frac{i^n}{(2\pi)^n n!} \mathrm{Tr}(\Omega^n) ,
\tag{70}
$$

which, because we are taking a trace, is rotation invariant. It is then enough to compute it at the north pole of the sphere $\vec{n} = (0,\dots,0,1)$ and then rotate it back to general position. In that case, since $d\vec{n} \cdot \vec{n} = 0$, we have $dn^{2n+1} = 0$. The computation greatly simplifies since

$$
\begin{aligned}
\Omega|_{\vec{n}=(0,\dots,0,1)} &= \frac{1}{16} \sum_{i,j=1}^{2n} (1+\gamma_{2n+1})\gamma_i\gamma_j(1+\gamma_{2n+1}) dn^i \wedge dn^j \\
&= \frac{1}{8} \sum_{i,j=1}^{2n} (1+\gamma_{2n+1})\gamma_i\gamma_j dn^i \wedge dn^j ,
\end{aligned}
\tag{71}
$$

because $\gamma_{2n+1}$ anti-commutes with all the remaining $\gamma$ matrices. We see that all we need to compute is

$$
\begin{aligned}
&\frac{i^n}{(2\pi)^n n! 2^{2n}} \sum_{i_1,j_1,\dots,i_n,j_n=1}^{2n} \frac{1}{2}\mathrm{Tr}\left[(1+\gamma_{2n+1})\gamma_{i_1}\gamma_{j_1}\cdots\gamma_{i_n}\gamma_{j_n}\right] dn^{i_1} \wedge dn^{j_1} \wedge \cdots \wedge dn^{i_n} \wedge dn^{j_n} \\
&= \frac{i^n}{2(2\pi)^n n! 2^{2n}} \sum_{i_1,j_1,\dots,i_n,j_n=1}^{2n} \varepsilon^{i_1 j_1 \dots i_n j_n} \mathrm{Tr}\left[(1+\gamma_{2n+1})\gamma_{i_1}\gamma_{j_1}\cdots\gamma_{i_n}\gamma_{j_n}\right] dn^1 \wedge \cdots \wedge dn^{2n} \\
&= (-1)^n \frac{(2n)!}{2(2\pi)^n n! 2^{2n}} \mathrm{Tr}\left[(1+\gamma_{2n+1})\gamma_{2n+1}\right] dn^1 \wedge \cdots \wedge dn^{2n} \\
&= (-1)^n \frac{(2n)!}{n!(2\pi)^n 2^{2n+1}} dn^1 \wedge \cdots \wedge dn^{2n} .
\end{aligned}
\tag{72}
$$

Observe that, by rotation invariance, the expression $dn^1 \wedge \cdots \wedge dn^{2n}$ is equivalent to, at a general point $\vec{n} \in S^{2n}$,

$$
\frac{1}{(2n)!} \sum_{i_1,i_2,\dots,i_{2n+1}=1}^{2n+1} \varepsilon_{i_1 \dots i_{2n+1}} n^{i_1} dn^{i_2} \wedge \cdots \wedge dn^{i_{2n+1}} ,
\tag{73}
$$

which is just the volume form of $S^{2n}$ with respect to the round metric $d\vec{n} \cdot d\vec{n}$, that has volume $(2^{2n+1}\pi^n n!)/(2n)!$. We can then use as a representative for the generator of top degree cohomology of $S^{2n}$

$$
\eta = \frac{1}{2^{2n+1} n! \pi^n} \sum_{i_1,i_2,\dots,i_{2n+1}=1}^{2n+1} \varepsilon_{i_1 \dots i_{2n+1}} n^{i_1} dn^{i_2} \wedge \cdots \wedge dn^{i_{2n+1}} ,
\tag{74}
$$

which satisfies $\int_{S^{2n}} \eta = 1$. In terms of $\eta$, we have

$$
\begin{aligned}
\frac{i^n}{(2\pi)^n n!} \mathrm{Tr}\,(\Omega^n) &= (-1)^n \frac{1}{(2\pi)^n 2^{n+1}} 2^{2n+1} \pi^n \eta \\
&= (-1)^n \eta\,.
\end{aligned}
\tag{75}
$$

In particular, the above equation implies that, since the quantum metric $g_{2n}$ is the round metric up to a scale factor, we can write $\eta = \frac{d\mathrm{vol}_{g_{2n}}}{\mathrm{vol}_{g_{2n}}(S^{2n})}$, and so we have

$$
\frac{i^n}{(2\pi)^n n!} \mathrm{Tr}\,(\Omega^n) = (-1)^n \frac{d\mathrm{vol}_{g_{2n}}}{\mathrm{vol}_{g_{2n}}(S^{2n})}\,.
\tag{76}
$$

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
