# Peer review of "Relating the topology of Dirac Hamiltonians to quantum geometry: When the quantum metric dictates Chern numbers and winding numbers"

_SciPost Physics, doi:SciPost Phys. 12, 018 (2022)_

## Round 3 · Referee Report · Anonymous (Referee 1) · 2021-7-27

Strengths

1- Original and greatly interesting idea for the quantum engineering and topological community 2- Very well written and highly comprehensible 3- Many different examples provided, showing that their construction applied to a variety of systems

Weaknesses

1-Eq. (1) and (2) in the introduction may look a bit complex for non experts, perhaps the authors can add a short summary in the intro about the meaning of Eq. (1) and (2) for a specific system

Report

The authors study the relation between two important concepts describing the geometry of electronic wavefunctions, the quantum metric and topological invariants given by the Berry curvature. These two objects has been at the center of a variety of novel effects in condensed matter systems, yet the relation between them has not been well established so far. The authors tackle this important problem, demonstrating that in a certain class of Dirac Hamiltonians of arbitrary dimension, a one-to-one relation between the quantum geometry and the topological invariants can be made. Interestingly, the findings of the authors suggest that topological invariants can be probed by dynamic susceptibility measurements that allow probing the quantum metric. Therefore, beyond the theoretical value of their work, this manuscript can be greatly relevant for future experiments in quantum matter.

The manuscript is well written and highly comprehensive, and the results presented are fully compatible with well-known results in the field. In particular, the authors explicitly demonstrate their idea for a variety of well-known topological states, including chiral phases and Chern insulators in 1,2,3,4 dimension. The manuscript provides a highly accessible demonstration of their original idea, and I believe that it would be of great interest to the theory community on topological and engineered quantum matter. For the reasons stated above, I strongly recommend the publication of their manuscript in Scipost Physics.

Requested changes

1- I would suggest that the authors can a short summary in the introduction of the meaning of Eq. (1) and (2), as they may look a bit complex for non expert readers

---

## Round 3 · Referee Report · Anonymous (Referee 3) · 2021-8-30

Strengths

1-generally comprehensive and readable write-up.
2-a mathematically interesting idea.
3-rather straightforward derivation.

Weaknesses

1- the idea, as far as presented in the paper, is only applicable to degenerate and unphysical class of Hamiltonians. 2-a great number of buzz words not directly related to the paper content in the abstract, introduction and conclusion confuse a potential reader and undermine the significance of the contribution. 3-no attempt to access the generatily of conclusions, which, in my opinion renders the paper a purely mathematical contribution.

Report

Dear all,
it's a first time I write a referee report for SciPost. Among other things, I was asked to check if this manuscript satisfies the acceptance criteria of SciPost. On a personal note, I find these criteria childish and silly. Moreover, their strict, literal and overall implementation would result - God forbids - in a quick and painfull death of physics as a branch of science. However, I do what I asked for.

This is a fine paper that address the Dirac Hamiltonians defined as linear superpositions of the Clifford algebra generators. It adresses in uniform manner arbitrariy space dimension as well as arbitrary dimension of the generators. While the members of Dirac Hamiltonian ensembles have been used as minimal models for various topological solids, an actual solid can never be described by Dirac Hamiltonian.

The main and, as far as a non-specialist can see, only achievement of the authors are two upper bonds on topological numbers in terms of "quantum"volume of the Brillion zone, expressed by Eqs. 1 and 2. This is an interesting idea that brings together two global characteristics of the geometry defined on periodical families of quantum states. It's a pity that it cannot be extended to any more physical situations.

I admit that the manuscript provides the comprehensive derivation of the main result for anybody who wishes to follow their exteremly formal outline, this derivation being decorated with a number of known examples.

This manuscript presents no discovery, breaktrhough, pathway with clear potential, neither a link between two research fields. As such, it does not satisfy the publication criteria of SciPost.

As mentioned, it is a fine paper with a solid scientific content. It may be published, for instance, in Physics Core, or any other decent journal. In my opinion, the best way to prepare the manuscript to this publication is to implement the changes I list under Requested Changes. By no means would I intrude the authors with an actual request: I just follow the formatting suggested by this particlular journal.

Requested changes

1- In the abstract: please remove 3 first sentences. They do not reflect the content of this paper yet may produce an unintended impression that those are results of this mere paper! 2- The whole 3-paragraph introduction seems generic and could suit almost any paper in the field. Strictly speaking, it is redundant and serves as a placeholder for citations. The real stuff begins in the subchapter "Scope..." More specific introduction would improve the presentation. 3- The concluding part consists of 7 sentences. From those, 2nd and 3rd is not directly related to the content of the work. The 6th and 7th might make sence but would require much more detailed explanations that, again, are not directly related to the research presented.

---

## Round 5 · Referee Report · Anonymous (Referee 1) · 2021-10-13

Strengths

  • The paper puts forward an interesting relation between Dirac Hamiltonians and quantum geometry, a topic of great interest for the condensed matter community

Weaknesses

  • All my previous comments have been address by the authors. No remaining weaknesses

Report

The authors have addressed the comments of the previous report and modified their manuscript accordingly. As elaborated in my previous report, I believe that their results are of great interest to the condensed matter community. I have also read in detail the report of Referee #3 and the response of the authors, and I believe that the authors have successfully addressed all the comments raised. Therefore, given all the points above, I strongly recommend the publication of their manuscript in Scipost Physics.

Requested changes

1- No changes requested

---

## Round 5 · Author Response

Dear Editors, Dear Referees,

We are glad to resubmit our manuscript entitled "Relating the topology of Dirac Hamiltonians to quantum geometry: When the quantum metric dictates Chern numbers and winding numbers" for your consideration as an Article in SciPost Physics.

First of all, we would like to thank both Referees for their reading of our work, and for their appreciations.

We have analyzed the two invited reports (#1 and #3) thoughtfully. The first report, which "strongly recommend(s) the publication in SciPost Physics" highlights the importance of our findings "for the quantum engineering and topological communities". This first report concludes by stating that our "manuscript can be greatly relevant for future experiments in quantum matter". In contrast, the second report (#3) contradicts those very same statements, by indicating that our idea "is only applicable to unphysical class of Hamiltonians", that our work is "a purely mathematical contribution" and that it is "a pity that it cannot be extended to any more physical situations". That same report (#3) nevertheless concludes that our manuscript is "a fine paper with a solid scientific content", which may be published in a "decent journal".

Above all, we strongly support the opinion of Referee #1 according to which the scope and results of our work are indeed relevant to experiments in quantum matter. Lattice systems described by a Dirac Hamiltonian (such as the emblematic 1D SSH model, the 2D Haldane model, the ideal 3D Weyl model, ... ) can be finely engineered in a broad class of physical settings (e.g. ultracold gases in optical lattices, photonics devices, electric circuits, ... ). Furthermore, these systems have been shown to be very well suited to measure quantum geometry [see, for instance, N. R. Cooper et al., Reviews of Modern Physics 91, 015005 (2019); T. Ozawa et al., Reviews of Modern Physics 91, 015006 (2019)]. In this sense, we strongly refute the statements of Referee #3 according to which our results are "only applicable to unphysical class of Hamiltonians" and that our work is "a purely mathematical contribution".

We hereby submit a revised version of our work, which takes the remarks and suggestions of the two Referees into account; see List of Changes below.

We hope that the Referees will be pleased by these revisions and that they will recommend publication of our work in the journal.

Yours sincerely,

The authors.

---

## Round 5 · List of Changes

Our changes are summarized below:

(1) Following the main criticism of Referee #3, the revised text better emphasizes the relevance of our work for ongoing experimental efforts. In particular, the text now explicitly refers to emblematic implementations of Dirac Hamiltonians in synthetic lattice systems, as well as to quantum-geometry measurements that have been recently performed in these systems.

(2) Following a suggestion of Referee #1, we now provide a "short summary in the introduction of the meaning of Eq. (1) and (2)". This reads [see below Eq. (2)]: "The relations presented in Eqs. (1) and (2) show that the volume of the Brillouin zone, as measured by the quantum metric, provides an upper bound to the topological invariants of generic Dirac Hamiltonians, in all dimensions. "

(3) Following a remark of Referee #3, we have included a whole new Section 6 dedicated to the implications of our metric-curvature relations for quantum metrology (an idea which was only briefly formulated in the original manuscript, and which, according to Referee #3 "require(d) much more detailed explanations").

(4) Following a remark of Referee #3, we have also clarified several statements in our concluding section, which the Referee #3 found "not directly related to the content of the work".

(5) We note that Referee #3 suggested to remove the three first sentences of our abstract because they "may produce an unintended impression that those are results of this mere paper". We have slightly revised the abstract so as to remove all possible ambiguity regarding the actual contributions of our work.

(6) Regarding the criticism of Referee #3 "The whole 3-paragraph introduction seems generic and could suit almost any paper in the field": We have decided to keep our introductory paragraphs in Section 1, because we believe that such an opening section can indeed be general before diving into more specific aspects; in particular, we believe that a general reader might appreciate this pluridisciplinary view on the quantum metric. We hope that the Referee #3 (who wrote "By no means would I intrude the authors with an actual request") will accept our choice of style.

---

## Editorial Decision

published